# A Systemic View of Carbohydrate Metabolism in Rice to Facilitate Productivity

**DOI:** 10.3390/plants10081690

**Published:** 2021-08-17

**Authors:** Woo-Jong Hong, Xu Jiang, Seok-Hyun Choi, Yu-Jin Kim, Sun-Tae Kim, Jong-Seong Jeon, Ki-Hong Jung

**Affiliations:** 1Graduate School of Biotechnology and Crop Biotech Institute, Kyung Hee University, Yongin 17104, Korea; hwj0602@khu.ac.kr (W.-J.H.); kangwuk97@khu.ac.kr (X.J.); lilchoi92@gmail.com (S.-H.C.); jjeon@khu.ac.kr (J.-S.J.); 2Department of Life Science and Environmental Biochemistry, Life and Industry Convergence Research Institute, Pusan National University, Miryang 50463, Korea; yjkim2020@pusan.ac.kr; 3Department of Plant Bioscience, Life and Industry Convergence Research Institute, Pusan National University, Miryang 50463, Korea; stkim71@pusan.ac.kr

**Keywords:** carbohydrate metabolism, microarray, crop, rice, productivity

## Abstract

Carbohydrate metabolism is an important biochemical process related to developmental growth and yield-related traits. Due to global climate change and rapid population growth, increasing rice yield has become vital. To understand whole carbohydrate metabolism pathways and find related clues for enhancing yield, genes in whole carbohydrate metabolism pathways were systemically dissected using meta-transcriptome data. This study identified 866 carbohydrate genes from the MapMan toolkit and the Kyoto Encyclopedia of Genes and Genomes database split into 11 clusters of different anatomical expression profiles. Analysis of functionally characterized carbohydrate genes revealed that source activity and eating quality are the most well-known functions, and they each have a strong correlation with tissue-preferred clusters. To verify the transcriptomic dissection, three pollen-preferred cluster genes were used and found downregulated in the *gori* mutant. Finally, we summarized carbohydrate metabolism as a conceptual model in gene clusters associated with morphological traits. This systemic analysis not only provided new insights to improve rice yield but also proposed novel tissue-preferred carbohydrate genes for future research.

## 1. Introduction

As the world’s population increases and arable land decreases year by year, food security has become one of the most serious problems faced by all countries [1]. Rice (*Oryza sativa* L.) is not only a model crop plant but also the main staple cereal that supplies nearly half of the world’s calorie consumption. Hence, improving its production is of great strategic significance for ensuring food security and sustainable agricultural development [2]. As a sessile and autophototrophic plant, rice generates carbohydrates by photosynthesis. These photoassimilates undergo a series of ordered metabolic processes and play a pivotal role in different developmental stages, including vegetative, reproductive, and ripening. Additionally, carbohydrate reserves in mature seeds provide the primary energy intake of mankind and contribute energy during its germination [3]. This source-sink coordination that runs through the entire plant life cycle reflects the importance of carbohydrate metabolism in rice productivity improvement.

Extensive research has provided evidence for the generation of more metabolic substrates by manipulating the potential of “source”, resulting in increased rice yield. For instance, *OsDWARF4* mutation showed an erect leaf phenotype that may enhance light capture for photosynthesis and finally lead to enhanced grain yield [4,5]. High grain yield was also observed in SNU-SG1 rice with the stay-green phenotype [6]. In addition to these, several attempts have been made to evaluate sugar transporters and key enzymes due to their vital role in carbohydrate metabolic processes. There are two main steps in sucrose translocation: phloem loading and unloading [7]. In the apoplastic loading model, one of the phloem loading steps, sucrose moves to the apoplasmic region and is loaded into phloem via Sucrose Transporters (SUTs) and Sugar Will Eventually be Exported Transporters (SWEETs) [8,9,10]. In post-phloem unloading, many studies have focused on sugar signaling after sucrose conversion into hexose by *Hexokinase* (*HXK*) family genes [11]. Moreover, the overexpression of *Grain Incomplete Filling 1* (*GIF1*), which encodes a cell wall invertase under the control of its native promoter, increases grain production [12]. Similarly, in maize, the constitutive expression of *Cell Wall Invertase* (*CWINV*) elevates grain yield and starch content [13].

Great progress has been made in this field. However, there has not been any big success until now, such as the green revolution caused by the discovery and application of semi-dwarf rice cultivars [14]. One explanation could be the failure to establish giant “sink” cultivars with rich spikelets due to the grain-filling ability that could not match a large yield capacity [15]. Another explanation is that carbohydrate metabolism has been oversimplified [8]; recently, there have been several reports regarding its complexity. For example, there is considerable heterogeneity in phloem loading and transport even in one species [16], and invertase inhibitors capping invertase exist [17]. An understanding of the systematic perceptions of carbohydrate metabolism for further applications is still very limited.

With the rise of bioinformatics and the establishment of high-throughput gene expression methods such as microarrays or next-generation sequencing technology, new technologies and methods have afforded systemic insights into various biological research fields. Recently, transcriptomic analyses of carbon partitioning during rice grain filling and the relationship between high temperature and grain filling have been carried out [18,19]. Despite the importance of systematic insights on carbohydrate metabolism in tissues related to morphological traits, transcriptome analysis has only been focused on a type of tissue and developmental processes.

To provide systemic insights into carbohydrate metabolism in rice, a transcriptomic dissection of carbohydrate metabolism-related genes retrieved from the Kyoto Encyclopedia of Genes and Genomes (KEGG) database [20] and the MapMan toolkit [21], which cover genome-wide biological pathways, was performed. After clustering genes with meta-expression profiles of anatomical samples, a functional enrichment analysis was performed, and the results were validated by quantitative reverse transcription-polymerase chain reaction (qRT-PCR). Finally, a conceptual model of carbohydrate metabolism to enhance crop yield was constructed. This research can shed light on carbon metabolism and provide candidate genes to enhance crop yield of rice and other species.

## 2. Materials and Methods

### 2.1. Integration of Carbohydrate Metabolism Annotation Data

Carbohydrate metabolism-related genes were collected according to the annotation of the MapMan toolkit (version 3.6.0RC1) [21] and KEGG database (retrieved on 10 April 2021 [20]. First, 266 carbon metabolism genes were selected from the KEGG database. Carbohydrate metabolism-related genes with MapMan bincodes from the MapMan toolkit were selected next. In total, 787 genes had MapMan bincodes (1: photosynthesis; 2: major CHO; 3: minor CHO; 4: glycolysis; 6: gluconeogenesis/glyoxylate cycle; 7: OPP; 8: TCA/org.trasnformation; 25: C1-metabolism; 34: transporters related to sugar or sucrose). Finally, 872 genes from the two data sources were selected, and 866 genes annotated by the Rice Genome Annotation Project (RGAP) [22] were chosen for further analysis.

### 2.2. Collection and Clustering of Microarray Data

Transcriptomic data were downloaded to analyze the anatomical expression patterns of carbohydrate metabolism-related genes. The data source mentioned was used in previous reports [23]. The detailed information is discussed below. For the analysis of anatomical expression profiles, anatomical data were retrieved from the Rice Oligonucleotide Array Database (ROAD) [24]. For heatmap analysis of cluster H genes, data were downloaded from RMEDB [25]. Multiple Experiment Viewer (MeV) is a widely used program for visualizing transcriptome data and performing statistical analysis [26]. MeV (version 4.9.0) was used to visualize the microarray data. For the dissection of transcriptome data, a k-means clustering (KMC) algorithm embedded in MeV was applied using the same method as with the identification of late pollen-preferred genes in rice [27]. Adobe Illustrator CS6 was used to edit the heatmap images.

### 2.3. Functional Classification via Literature Search

To find the previously characterized functional roles of the 866 carbohydrate metabolism-related genes in anatomical clusters, the Overview of Functionally characterized Genes in Rice Online (OGRO) database (http://qtaro.abr.affrc.go.jp/ogro/table (accessed on 14 April 2021)) was used [28]. Information for 1949 functionally characterized genes is available in this database. As in a previous study [29], information on the 866 genes was parsed, and data were summarized using Excel 365 (version 16.0.14228.20158). Count numbers for the characterized genes were visualized using R Studio (version 1.4.1106) and ggplot2 R package (version 3.3.3) [30]. The detailed, functionally characterized gene information of the 866 carbohydrate metabolism-related genes is listed in Appendix A.

### 2.4. Gene Ontology (GO) Enrichment Analysis

GO enrichment is commonly used to interpret the functional roles of large-scale transcriptomic data [31]. This study used the ROAD to find the GO terminology for each cluster (http://ricephylogenomics-khu.org/road/go_analysis.php, temporary homepage for updating (accessed on 7 May 2021)). To perform the GO enrichment analysis, the following criteria were applied: query number > 2, hyper *p* < 0.05, and fold enrichment value (query number/query expected number) > 2 [32]. Significant GO terms and integrated cluster information were selected from the transcriptome data analysis with each selected GO term. Finally, these data were visualized via R Studio (version 1.4.1106) and ggplot2 R package (version 3.3.3).

### 2.5. KEGG Enrichment Analysis

KEGG enrichment analysis was performed using R Studio and the clusterProfiler package [33]. To use the enrichKEGG function in this package, input data consisting of cluster information and Rice Annotation Project Database ID (https://rapdb.dna.affrc.go.jp/ (accessed on 7 May 2021)) [34] were used. In addition, “dosa” was chosen as the organism code, and the results were filtered out by applying an adjusted *p*-value cutoff < 0.05. For the visualization of the results, the dot-plot function in the package was used, and the figure was modified with the ggplot2 package (version 3.3.3).

### 2.6. RNA Extraction and qRT-PCR

To isolate RNA, plants were grown in a paddy field condition, as reported previously [35]. Samples were immediately frozen in liquid nitrogen, and total RNA was isolated using a TRIzol reagent (Invitrogen, Waltham, MA, USA) combined with an RNase Plant Mini Kit (Qiagen, Hilton, Germany; http://www.qiagen.com (accessed on 7 May 2021)) and DNase treatment. First-strand cDNA was synthesized using the SuPrimeScript RT Premix (with oligo(dT), 2×; GeNet Bio, Daegu, Korea). A qRT-PCR was performed, as reported previously [36]. For the *gori* knockout mutant, anthers from a paddy field-grown plant were collected to extract RNA. All primers used in this study are listed in Appendix A.

### 2.7. Construction of the Conceptual Carbohydrate Metabolism Model

To generate a conceptual model focusing on the source-sink communication pathway, four key enzymes were selected: invertase (INV), sucrose synthesis (SUS), sucrose transporter (SUT), and hexokinase (HXK). Cluster information was then integrated by indicating an organ/tissue-preferred expression pattern: clusters A and B for leaf, cluster E for root, cluster H for pollen, cluster I for grain, and cluster J for ubiquitous expression patterns. Finally, the regulatory network between *GORI* and three cluster H genes was incorporated after adding qRT-PCR data between the *gori* knockout mutant and wild type anthers.

Rice plant images were downloaded from the International Rice Research Institute (https://www.flickr.com/photos/ricephotos/albums/72157643341257395) (accessed on 7 May 2021)), and the images were arranged using Adobe Illustrator CS6 (version 16.0.0).

## 3. Results

### 3.1. Identification of Genome-Wide Candidate Genes Related to Carbohydrate Metabolism

The MapMan toolkit and the KEGG database are useful information sources for the functional annotation of large-scale genes [20,21]. These two data sources were used to retrieve reliable carbohydrate metabolism-related genes. First, 266 genes involved in carbon metabolism pathways were found in the KEGG database. These genes were also searched in the MapMan toolkit, and genes in bincodes related to carbohydrate metabolism were identified: photosynthesis, major CHO, minor CHO, glycolysis, gluconeogenesis/glyoxylate cycle, OPP, TCA/organic acid transformation, and C1-metabolism. In addition to these bincodes, genes related to sugar or sucrose transport were added. Finally, 872 genes were collected using two public annotation sources. Because most expression data were available with locus IDs from the RGAP website (http://rice.plantbiology.msu.edu/ (accessed on 15 July 2021)), further analysis was performed on 866 candidate genes with RGAP locus IDs (Figure 1; Appendix A).

### 3.2. Functional Analysis of the Characterized Carbohydrate Metabolism-Related Genes

To analyze the functional significance of the 866 carbohydrate metabolism-related genes, functionally characterized genes were searched among them. To do this, the OGRO website was used [28]. Information of 1949 functionally characterized genes was then retrieved and classified according to major functional categories such as physiology, morphology, tolerance, or resistance. The functionally characterized roles for 76 of the 866 genes, including duplicate information about one locus, were identified (Figure 2). In the physiology category, eating quality was related to 22 genes, source activity was related to 19 genes, and flowering was related to three genes. In the morphology category, dwarf was related to five characterized genes, seed was related to four genes, and culm/leaf and root were related to two genes. Finally, regarding tolerance or resistance, salinity tolerance was related to five genes and cold and drought tolerance was related to two genes. As expected, the most frequently characterized functional category associated with carbon metabolism was eating quality in the physiology category, followed by source activity (Table 1). This result indicates that 866 carbohydrate genes might be useful candidates for enhancing the grain yield of rice associated with eating quality and source activity.

### 3.3. Anatomical Dissection of Carbohydrate Metabolism-Related Genes via Meta-Expression Analysis

To assess the functional roles of the 866 carbohydrate metabolism-related genes, meta-anatomical expression profiles consisting of 983 rice Affymetrix array anatomical sample data were first used [23]. Using the KMC algorithm, 729 genes with probes on the Affymetrix array were grouped into 11 anatomical clusters (Figure 3a; Appendix A). This analysis could not be performed for 137 genes without probes on the Affymetrix array. Based on this analysis, carbohydrate metabolism-related genes may be involved in diverse morphological or physiological traits. For example, clusters A and B are closely associated with leaf and shoot development, cluster E is associated with root, clusters G and H are associated with pollen, and cluster I is associated with grain. In addition, cluster J, with ubiquitous expression patterns, might be related to the housekeeping function.

### 3.4. Functional Comparison and Enrichment Analysis of 11 Anatomical Clusters

The distribution of functionally characterized genes among clusters was searched to identify the relationships between anatomical expression patterns and the 11 clusters. Subsequently, 13 source activity genes were enriched in cluster A, which showed a leaf-preferred expression pattern. Similarly, 15 eating quality genes were in cluster I, which showed a grain-preferred expression pattern. Other clusters did not show a strong correlation between known functions and featured expression patterns (Appendix A).

Enrichment analysis of functional groups was also performed for each of the 11 anatomical clusters. To do this, GO and KEGG enrichment analyses were conducted. As a result, four GO terms associated with photosynthesis were enriched in cluster A: reductive pentose phosphate cycle (GO: 0019253), photosynthesis-light harvesting (GO: 0009765), photosynthesis (GO: 0015979), and electron transport chain (GO: 0022900). In cluster I, there were no photosynthesis-related GO terms. Instead, there were four GO terms related to sugar metabolism or biosynthesis: sucrose metabolic process (GO: 0005985), starch biosynthetic process (GO: 0019252), glucan biosynthetic process (GO: 0009250), and glycogen biosynthetic process (GO: 0005978; Figure 3b).

Consistent with the GO enrichment analysis results, KEGG enrichment also showed that photosynthesis pathways were enriched in cluster A, and starch and sucrose metabolism pathways were enriched in cluster I (Figure 3c). These results suggest that metabolic pathways might be further dissected by diverse developmental processes. Assigning expression patterns to each of the clusters will be useful for further functional analysis.

### 3.5. In Silico and In Vitro Expression Verification of Tissue-Preferred Genes

To validate the functional significance of gene clusters according to anatomical expression patterns, three genes (*Os10g26740*, *Os02g06540*, and *Os10g08022*) in cluster H associated with anther and pollen development were selected: sucrose transporter (*OsSUT3*), monosaccharide transporter, and fructose-bisphosphate aldolase isozyme, respectively. In silico analysis revealed that all these genes showed anther/pollen-preferred expression patterns (Figure 4a). This expression pattern was further confirmed by qRT-PCR, with samples in six tissues/organs (Figure 4b) matching the dissected model. Recently, a defect in the *GORI* gene changed the distribution of pectin in germinated pollen tubes and eventually led to the male sterile phenotype [35]. Thus, qRT-PCR was performed to analyze their regulatory roles by *GORI* in the *gori* mutant. Interestingly, all three genes were significantly downregulated in the *gori* mutant than in the wild-type (Figure 4c). It was speculated that these transporters and carbon metabolism-related enzymes regulated by *GORI* might be involved in the pollen tube growth process.

### 3.6. Construction of a Conceptual Carbohydrate Metabolism Model

In this section, a conceptual carbohydrate metabolism model that will help improve crop yield, as constructed previously for the rice endosperm, is proposed [89] (Figure 5). Notably, several studies fit well with this model. In pollen cluster H, *OsHXK10* is involved in anther dehiscent and pollen germination [90]. In addition, *CWINV3* mutation caused male sterility [91]. Regarding grain cluster I, *SUS3* overexpression increased cell wall polysaccharide deposition, resulting in enhanced biomass saccharification [92].

This model, consistent with several functionally characterized genes, will be useful in providing guidelines for the spatial manipulation of carbohydrate metabolism-related genes in order to enhance crop yield.

## 4. Discussion

Improvement in crop yield is becoming more urgent due to the need to supply nearly 10 billion people. To maintain food security, many studies on carbohydrate metabolism in rice have been performed. However, these studies mostly focused on the source-sink mechanism and specific temporal samples such as the grain-filling stages. To complement this uneven interpretation of carbohydrate metabolism and provide new insights on rice productivity to enhance research associated with carbon metabolism, 866 carbohydrate metabolism-related genes were systemically dissected into 11 clusters according to meta-anatomical expression data (Figure 3; Appendix A).

As mentioned above, a functionally characterized gene search and functional group enrichment analyses indicated that most functionally characterized carbohydrate metabolism-related genes were involved in the source-sink mechanism (Figure 2). Along with the results, this analysis showed some clusters showing root-, pollen-, and seed-preferred expression patterns (clusters E, H, and I). From among these, the expressions of three pollen-preferred carbon metabolism-related genes were confirmed, supporting the reliability of meta-anatomical expression data in this study (Figure 4).

Improving the seed setting rate through carbohydrate metabolism is a means of elevating rice productivity. One study reported that glycolysis could regulate pollen tube polarity in *Arabidopsis* [93]. Similarly, a recently characterized gene (*GORI*) involved in late pollen development in rice showed its connectivity to carbohydrate metabolism such as less pectin staining in the *gori* pollen tube [35]. Interestingly, three genes in cluster H showed significantly reduced expression when *GORI* was knocked out (Figure 4c). This result suggested that cluster H could be a suitable research candidate for further productivity improvement in the context of carbohydrate metabolism underlying late pollen development in rice.

Grain filling is also an important factor for rice yield increase [12]. In this analysis, the seed-preferred cluster I includes *SUS3* and *SUS4*, characterized as grain filling-related genes [94]. In addition, SWEET and glutamine synthetase, which were excluded in the analysis due to the limitations of the data source for functional categorization, play an important role in the grain-filling process related to sucrose transport and nitrogen metabolism, respectively [95,96,97]. Furthermore, when a hierarchical clustering of the *SWEET* genes with the 11 anatomical clusters was performed, *SWEET11* and *SWEET15* were close to cluster I (data not shown). Jointly, carbohydrate genes in cluster I could be useful genetic resources for further investigation of the rice grain-filling process and other metabolic processes.

Moreover, crop productivity can be affected by various factors, including environment, fertilizer, soil conditions, and even rhizobiome composition [2,98,99]. In particular, the interaction between crop root and the rhizobiome is related to root exudates, including amino acids, secondary metabolites, and carbohydrates [100]. Although the investigation of root exudates is understudied until now, the root-preferred cluster E could be an appropriate target for further studies on carbohydrate metabolism for generating exudates in the root.

## 5. Conclusions

This study aimed to improve the overall understanding of carbohydrate metabolism, which could provide some unknown clues for increasing rice productivity. For this, 866 carbohydrate metabolism-related genes were integrated into meta-anatomical expression data, and the significance of each cluster was shown using the functionally characterized roles in each cluster. Through an integrated analysis, carbohydrate metabolism-related genes were systemically dissected into 11 tissue-preferred clusters. Further functional enrichment analysis showed that two clusters (A and I) were strongly associated with source- and sink-preferred roles, respectively. In addition, the expression patterns of three pollen-specific cluster H genes were provided as examples of the reliability of the analysis. Furthermore, the reduced expression of the three cluster H genes in the *gori* mutant suggested that the tissue-preferred clusters could be suitable targets for further investigation. Collectively, a conceptual carbohydrate metabolism model summarizing the results was constructed, and it provided holistic insights on carbohydrate metabolism and suggested suitable candidates for improving crop productivity beyond source-sink mechanism-focused research.

## Figures and Tables

**Figure 1 plants-10-01690-f001:**
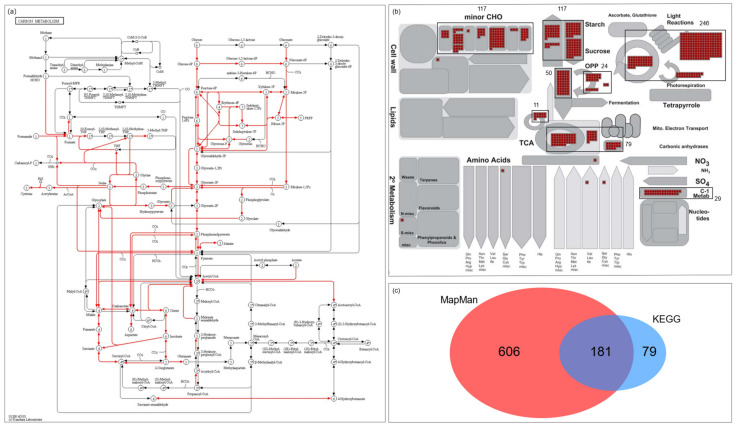
Schematic diagram summarizing 866 carbohydrate metabolism-related genes retrieved from the KEGG database and the MapMan toolkit. In the KEGG database, carbon metabolism pathway genes were selected and applied to the MapMan toolkit to find associated MapMan annotations. (**a**) There were eight bincodes related to 266 KEGG genes and with multiple members. (**b**) The bins with a black box and the total number of whole elements in each bin were indicated. Therefore, 681 MapMan genes associated with carbohydrate metabolism were selected. In addition, 106 carbohydrate transporters annotated in MapMan were included. A total of 787 genes from the MapMan toolkit and 266 from the KEGG database were collected. (**c**) Finally, 872 carbohydrate metabolism-related genes were collected. For further analysis, 866 genes with RGAP locus information were used. The detailed information of the 866 genes, including 181 overlapped genes, is listed in Appendix A.

**Figure 2 plants-10-01690-f002:**
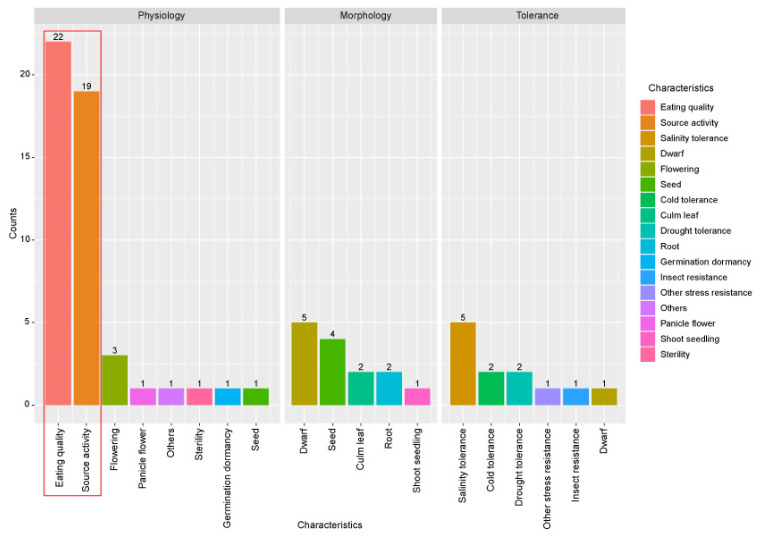
Distribution of functionally characterized genes from 866 carbohydrate metabolism-related genes according to phenotype. A literature search was performed to analyze the functional significance of the 866 carbohydrate metabolism-related genes. Functionally characterized gene information was retrieved from the OGRO database. To visualize the results of the 76 functionally characterized genes indicated in Table 1, three categories of major characteristic information from the OGRO database were used: physiology, morphology, and tolerance/resistance. The number of characterized genes was counted according to the minor characteristics within three major categories: physiology, morphology, and tolerance. The red box indicates the most enriched minor characteristics of the functions associated with the 76 functionally characterized genes.

**Figure 3 plants-10-01690-f003:**
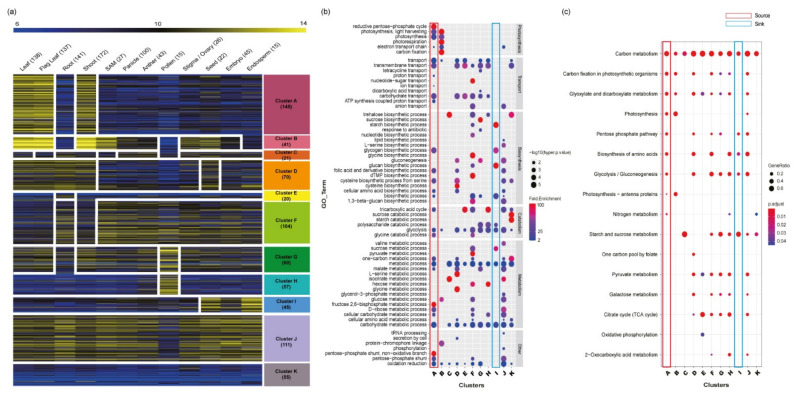
Dissection of the 866 carbohydrate metabolism-related genes using meta-anatomical expression profiles and functional enrichment analysis of 11 anatomical clusters. (**a**) Using large-scale microarray data, carbohydrate metabolism-related genes were visualized and dissected. The heatmap of the anatomical expression profiles of carbohydrate metabolism-related genes was grouped into 11 clusters via KMC clustering methods. The numbers after the name of each tissue/organ indicate the sample size, and the numbers below each cluster indicate the number of genes in the cluster. (**b**) GO enrichment analysis of 11 anatomical clusters. The GO enrichment assay revealed the characteristics of each cluster. GO terms were classified according to biological process GO terms. Dot color indicates the fold enrichment value (the blue color is 2, which is the minimum cutoff to select a significant fold enrichment value, and the red color indicates a higher fold enrichment value), and dot size indicates statistical significance (-log10(hyper *p*-value) is used, and a larger dot size means more significance). (**c**) KEGG enrichment analysis of 11 anatomical clusters. The enriched KEGG pathway indicated with the dot size represents the ratio of the selected genes to the total genes in the pathway, and the dot color illustrates the adjusted *p*-value. The numbers below the clusters indicate the number of mapped genes to each pathway. In addition, in GO and KEGG enrichment analyses, the source- and sink-related clusters are highlighted as red and blue boxes, respectively.

**Figure 4 plants-10-01690-f004:**
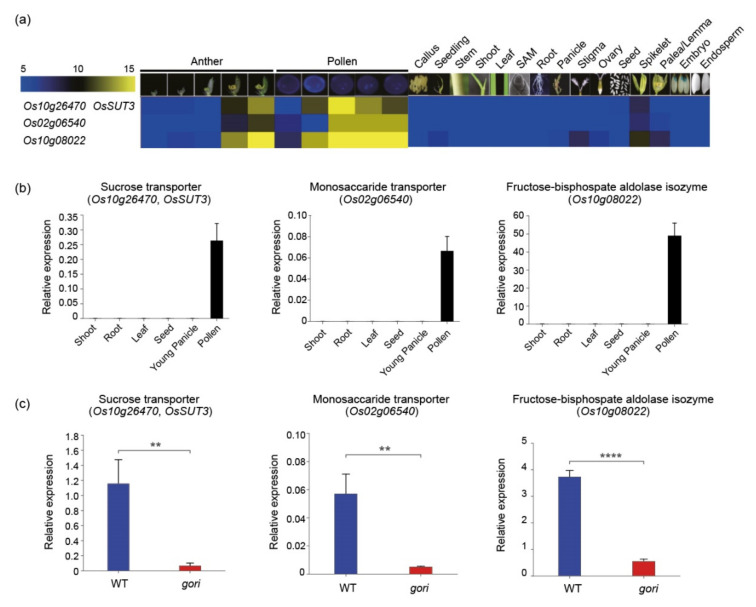
Validation of the dissected model and expression analysis of three carbohydrate metabolism-related genes in cluster H. (**a**) Heatmap analysis of three cluster H genes (*Os10g26740*, *Os02g06540*, and *Os10g08022*). Numeric values indicate an average of the normalized log_2_ intensity values. (**b**) Expression profiles of the three pollen-preferred genes based on qRT-PCR in various rice tissues: shoot, root, leaf, seed, young panicle, and pollen. (**c**) The expression of three pollen-preferred carbohydrate genes was significantly downregulated in the *gori* mutant compared to the wild-type plants. *OsUbi5* (*Os01g22490*) was used as an internal control. There were three biological replicates from the performed *t*-test on independent samples, with Bonferroni correction. ** 0.001 < *p* ≤ 0.01; **** *p* ≤ 0.0001.

**Figure 5 plants-10-01690-f005:**
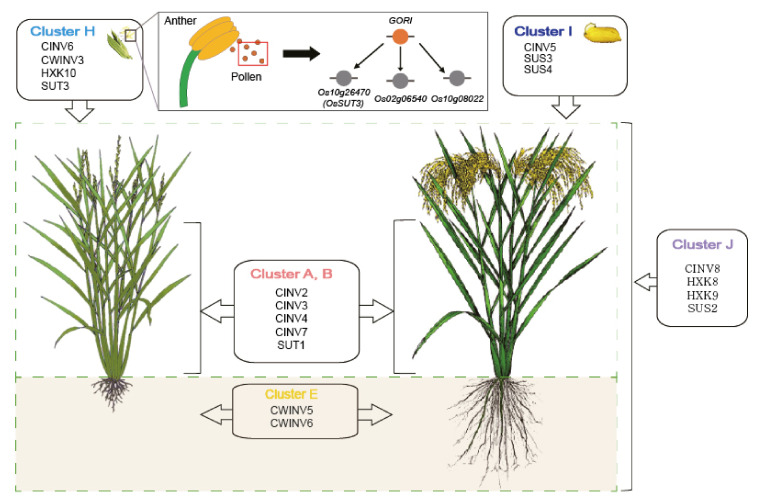
Conceptual carbohydrate model integrated with cluster information. A conceptual model was constructed by summarizing dissected carbohydrate metabolism-related genes according to the anatomical expression pattern. This model indicates the clusters associated with key enzymes (INVs, SUTs, SUSs, and HXKs) for source-sink communication. Clusters A and B are associated with leaf and flag leaf, cluster E is associated with root, cluster H is associated with anther and pollen, cluster I is associated with grain, and cluster J is associated with the whole rice plant based on ubiquitous expression patterns. The *GORI* regulatory model for pollen tissue was also combined.

**Table 1 plants-10-01690-t001:** Summary of functionally characterized genes from the 866 carbohydrate metabolism-related genes.

LOC_id	Symbol	Character_major ^1^	Character_minor	Method ^2^	Detailed Functions	Reference
Os02g44230	OsTPP1	R or T	Cold tolerance	OX	Cold and salinity tolerance	[37]
Os05g44210	OsTPS1	R or T	Cold tolerance	OX	Cold, drought, and salinity tolerance	[38]
Os01g64660	moc2	MT	Culm leaf	KO	Tiller bud outgrowth; tillering	[39]
Os12g44380	ossut2	MT	Culm leaf	KO	Sugar export; growth retardation	[40]
Os05g44210	OsTPS1	R or T	Drought tolerance	OX	Cold, drought, and salinity tolerance	[38]
Os06g36560	OsMIOX	R or T	Drought tolerance	OX	Drought tolerance	[41]
Os03g07480	OsSUT1	MT	Dwarf	KD	Dwarfism; flowering time	[42]
Os04g56320	OsrcaA2	MT	Dwarf	KD/OX	Dwarfism	[43]
Os07g05820	OsGLO4	R or T	Dwarf	KD	Growth inhibition	[44]
Os11g05110	OsPK1	MT	Dwarf	KO	Elongation of uppermost internode; dwarfism	[45]
Os11g05110	ospk1	MT	Dwarf	KO	Dwarfism; seed color; internode color; regulation of gibberellin and ABA biosynthesis	[46]
Os12g44380	ossut2	MT	Dwarf	KO	Sugar export; growth retardation	[40]
Os01g44220	osagpl2	PT	Eating quality	KO	Starch biosynthesis	[47]
Os01g55540	OsAAT2	PT	Eating quality	OX	Seed amino acid and protein content	[48]
Os02g52710	Amy1A	PT	Eating quality	KD	Seed starch content; high temperature-triggered grain chalkiness	[49]
Os03g03720	OsGAPDHB	PT	Eating quality	KD	Fragrance rice	[50]
Os03g09250	RINO1	PT	Eating quality	KD	Seed phytic acid content	[51]
Os03g09250	RINO1	PT	Eating quality	KD	Seed phytic acid content	[52]
Os03g52760	lpaN15-186	PT	Eating quality	KO	Seed phytic acid content	[53]
Os03g55090	pho1	PT	Eating quality	KO	Seed starch content; grain maturation	[54]
Os05g32710	OsISA2	PT	Eating quality	OX	Seed starch content	[55]
Os05g33570	flo4	PT	Eating quality	KO	Seed protein and lipid content	[56]
Os06g04200	wx	PT	Eating quality	NV	Seed amylose content	[57]
Os06g06560	OsSSI	PT	Eating quality	KO	Seed amylopectin content	[58]
Os06g12450	SSIIa	PT	Eating quality	KD	Chalky kernel; amylose content	[59]
Os06g12450	ALK	PT	Eating quality	NV	Gelatinization temperature; gel consistency	[60]
Os08g09230	SSIIIa	PT	Eating quality	KD	Chalky kernel; amylose content	[59]
Os08g09230	flo5	PT	Eating quality	KO	Seed starch content	[61]
Os08g09230	ss3a	PT	Eating quality	KO	Seed starch content	[62]
Os08g25734	osagps2	PT	Eating quality	KO	Seed starch content	[47]
Os08g40930	OsISA1	PT	Eating quality	OX	Seed starch content	[55]
Os09g28400	Amy3A	PT	Eating quality	KD	Seed starch content; high temperature-triggered grain chalkiness	[49]
Os09g28420	Amy3B	PT	Eating quality	KD	Seed starch content; high temperature-triggered grain chalkiness	[49]
Os09g29404	isa3	PT	Eating quality	KO	Seed starch content	[63]
Os01g72090	se13	PT	Flowering	KO	Photoperiodic response	[64]
Os02g34560	Oscyt-inv1	PT	Flowering	KO	Root cell development; flowering time; fertility	[65]
Os03g07480	OsSUT1	PT	Flowering	KD	Dwarfism; flowering time	[42]
Os02g52710	AmyI-1	PT	Germination dormancy	KD/OX	Starch degradation; seed germination; seedling growth	[66]
Os03g07480	OsSUT1	R or T	Insect resistance	Others	Upregulation by aphid feeding; transfer sucrose	[67]
Os02g47020	SBPase	R or T	Other stress resistance	OX	Photosynthetic ability under high-temperature condition	[68]
Os02g01150	OsHPR1	PT	Others	KD/OX	Photorespiratory pathway	[69]
Os05g50380	LSU3	Others	Others	KO	Culm starch content	[70]
Os01g69030	OsSPS1	PT	Panicle flower	KO	Pollen germination through sucrose synthesis	[71]
Os02g34560	Oscyt-inv1	MT	Root	KO	Root cell development; flowering time; fertility	[65]
Os12g44380	ossut2	MT	Root	KO	Sugar export; growth retardation	[40]
Os02g17500	OsGMST1	R or T	Salinity tolerance	KD	Salinity tolerance	[72]
Os02g44230	OsTPP1	R or T	Salinity tolerance	OX	Cold and salinity tolerance	[37]
Os03g07480	OsSUT1	R or T	Salinity tolerance	KD	Salinity tolerance	[73]
Os05g44210	OsTPS1	R or T	Salinity tolerance	OX	Cold, drought, and salinity tolerance	[38]
Os08g03290	OsGAPC3	R or T	Salinity tolerance	OX	Salinity tolerance	[74]
Os01g44220	OsAGPL2	MT	Seed	KO	Glassy/vitreous, shrunken grain	[75]
Os03g55090	pho1	MT	Seed	KO	Seed starch content; grain maturation	[54]
Os04g33740	GIF1	MT	Seed	NV	Grain filling; grain size	[12]
Os08g25734	OsAGPS2b	PT	Seed	KO	Seed weight; starch content; AGPase activities from developing endosperms of the seed	[75]
Os11g05110	ospk1	MT	Seed	KO	Dwarfism; seed color; internode color; Regulation of gibberellin and ABA biosynthesis	[46]
Os02g52710	AmyI-1	MT	Shoot seedling	KD/OX	Starch degradation; seed germination; seedling growth	[66]
Os01g11054	Osppc4	PT	Source activity	KD	Ammonium assimilation in leaves	[76]
Os01g64660	oscfbp1	PT	Source activity	KO	Photosynthetic sucrose biosynthesis; growth retardation	[77]
Os01g64960	qNPQ1-2	PT	Source activity	NV	Nonphotochemical quenching capacity; protection from photoinhibition	[78]
Os01g64960	PsbS	PT	Source activity	KD/OX	Nonphotochemical quenching capacity; photosynthetic rate in fluctuating light conditions	[79]
Os02g32660	BE2b	PT	Source activity	Others	Starch biosynthesis in endosperm; amylopectin biosynthesis; branch formation	[80]
Os03g57220	OsGLO1	PT	Source activity	KD	Photorespiration	[81]
Os04g56320	rca	PT	Source activity	KD	Rubisco activity	[82]
Os05g40180	OsSTN8	PT	Source activity	KO	Photosystem II repair during high light illumination	[83]
Os06g06560	SS1	PT	Source activity	Others	Starch biosynthesis in endosperm; chain elongation	[80]
Os06g51084	BE1	PT	Source activity	Others	Starch granule binding; amylopectin structure	[80]
Os07g05820	OsGLO4	PT	Source activity	KD	Rubisco activation; photosynthesis rate	[44]
Os08g45190	PGR5	PT	Source activity	KD	Photosynthetic capacity	[84]
Os10g37180	OsGDCH	PT	Source activity	KD	Leaf senescence induced by reactive oxygen species	[85]
Os12g17600	rbcS	PT	Source activity	KD	Rubisco content; photosynthetic capacity	[86]
Os12g17600	OsRBCS2	PT	Source activity	KD	Rubisco content	[87]
Os12g17600	rbcS	PT	Source activity	KD/OX	Rubisco content; photosynthetic capacity	[88]
Os12g19381	OsRBCS5	PT	Source activity	KD	Rubisco content	[87]
Os12g19470	OsRBCS4	PT	Source activity	KD	Rubisco content	[87]
Os12g44380	ossut2	PT	Source activity	KO	Sugar export; growth retardation	[40]
Os02g34560	Oscyt-inv1	PT	Sterility	KO	Root cell development; flowering time; fertility	[65]

^1^ Major functional categories: R or T, resistance or tolerance; MT, morphological trait; PT, physiological trait. ^2^ Methods used to study: OX, overexpression; KO, knockout; KD, knockdown; NV, natural variation.

## Data Availability

The expression data presented in this study are available in the Appendix A.

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
