# Peer review of "A Systemic View of Carbohydrate Metabolism in Rice to Facilitate Productivity"

_plants, 2021, doi:10.3390/plants10081690_

Round 1

Reviewer 1 Report

The manuscript by Hong et al. presents the functionally characterized carbohydrate genes regarding developmental growth and yield-related traits in rice plants. They constructed and proposed a conceptual carbohydrate metabolism model to understand rice metabolism and to get related clues for increasing yield. Two major metabolic databases and gene annotations were used to do it. The constructed model also supports recent results for grain filling. I understand that such a work can contribute to our understanding of the highly complex regulation of carbohydrate metabolism in rice. However, I had some major comments and also felt that this study was too preliminary.

Major comments:

  1. Aims of this study

One of lacking information is the aims of this study.

  1. Materials and Methods

2-1) The version of MapMan and KEGG annotation is lacking.

2-2) Please cite the latest paper regarding all the databases and software used. For example, RGAP.

2-3) There was no description about adjusting p-values for multiple testing problems in GO enrichment and KEGG pathway analyses.

  1. Figures and Tables

3-1) The image resolution of all the figures is too low. For example, we cannot see any pathways in Figure 3b.

3-2) Figure 1c: What are 181 overlap genes?

  1. Results and Discussion

4-1) Detail construction procedure of a Conceptual Carbohydrate Metabolism Model is lacking.

4-2) Discussion part is too poor. What is the novelty of this study?

4-3) Similar to transporters like SWEETs, it is known that glutamine synthetase and its isoforms in rice are important for grain development. Do you have any ideas?

Author Response

Comments 1: Aims of this study: One of lacking information is the aims of this study.

Response to comment 1: We appreciate your careful comment; the purpose of this study is to provide systemic insight about carbohydrate metabolism using transcriptome data. By clustering and characterizing the carbohydrate metabolism genes, we could identify tissue preferred genes like cluster E, H, and I. Especially, in case of the pollen preferred expressed cluster H genes, it could be important candidates related to pollen germination or pollen tube growth which is an important process for rice seed setting rate. To clarify this point, we added the aim at lines 25-27, and 73-84. 

Comments 2: The version of MapMan and KEGG annotation is lacking.

Response to comment 2: To address this comment, we added version information about those two databases (Lines 88-89).

Comments 3: Please cite the latest paper regarding all the databases and software used. For example, RGAP.

Response to comment 3: Thank you for your comment, we updated citation of the references for used database and software during revision (Lines 88, 95 and 133). 

Comments 4: There was no description about adjusting p-values for multiple testing problems in GO enrichment and KEGG pathway analyses.

Response to comment 4: We adapted criteria including hyper p-value threshold as previously reported (Jung et al., 2008) In the reference paper, hyper p-value 0.05 is statistically meaningful FDR p-value 1e-6. In the case of KEGG pathway analysis, we used adjusted p-value using FDR method embedded in the clusterProfiler package. Detailed description about this point is added on the revised manuscript (Lines 126, and 134). 

Comments 5: The image resolution of all the figures is too low. For example, we cannot see any pathways in Figure 3b.

Response to comment 5: To address this comment, We revised all the figures with enhanced resolution (Figures 1-5).

Comments 6: Figure 1c: What are 181 overlap genes?

Response to comment 6: The 181 overlapping genes are listed in both data sources (MapMan and KEGG database). To provide detailed information, we add its information in the supplementary file (Table S3; Lines 177-178).

Reviewer 2 Report

Interesting paper regarding carbohydrate metabolism in rice to facilitate productivity.

However, the following must be done, in order to improve the shape and the content of the manuscript.

English must be carefully revised. Please avoid using the personal manner of addressing “we”, “our”, (it is annoying so many “we”) and use the impersonal one; the text will sound much more professional. I.e. just in a single paragraph L76-80 of 5 lines, ‘we” is used 4 times! (L76 we then clustered – incorrect grammar, please reshape; we performed L 77; we proposed L78; we hope L79), replace them with It was clustered, it was performed, we hope – is not a scientific language. Please revise the entire manuscript in this regard, beginning with the Abstract. It is very clear that is “yours” paper.

No need to underline the first letter of the words in L74, L108, L115, etc. Please remove the underline in the entire manuscript.

In 2. Materials and Methods section, I recommend a subsection to be added, related to the criteria of papers selection. In this regard, a PRISMA flow chart is recommended, according to Page, M.J.; et al. The PRISMA 2020 statement: An updated guideline for reporting systematic reviews. Journal of Clinical Epidemiology 2021, 134, 178-189, doi:10.1016/j.jclinepi.2021.03.001.; Page, M.J.; et al. Updating guidance for reporting systematic reviews: development of the PRISMA 2020 statement. Journal of Clinical Epidemiology 2021, 134, 103-112, doi:10.1016/j.jclinepi.2021.02.003.

L112. EXCEL which version?

Discussion part must be extended, it is much too short. As the title of the paper mentions “… to Facilitate Productivity” in the Discussion section it must be mentioned a paragraph related to the ways of increasing production taking into account the soil properties, crop rotation, climatic aspects and the use of fertilizers. Recent/ very recent data are provided by the following references (please check them): Bungau S.; et al. Expatiating the impact of anthropogenic aspects and climatic factors on long term soil monitoring and management. Environ Sci. Pollut. Res. 2021, 30528-30550. https://doi.org/10.1007/s11356-021-14127-7 ;Samuel A.D.; et al.  Effects of long term application of organic and mineral fertilizers on soil enzymes, Rev. Chim. 2018, 69(10), 2608-2612. https://doi.org/10.37358/RC.18.10.6590    ; Samuel, A.D.; et al.  Effects of liming and fertilization on the dehydrogenase and catalase activities, Rev. Chim. 2019, 70(10), 2019, 3464-3468. https://doi.org/10.37358/RC.19.10.7576 ; Gitea M.A., et al.  Orchard management under the effects of climate change: implications for apple, plum and almond growing, Environ. Sci. Pollut. Res. 2019, 26(10), 9908-9915. https://doi.org/10.1007/s11356-019-04214-1

Author Response

Comment 1: English must be carefully revised. Please avoid using the personal manner of addressing “we”, “our”, (it is annoying so many “we”) and use the impersonal one; the text will sound much more professional. I.e. just in a single paragraph L76-80 of 5 lines, ‘we” is used 4 times! (L76 we then clustered – incorrect grammar, please reshape; we performed L 77; we proposed L78; we hope L79), replace them with It was clustered, it was performed, we hope – is not a scientific language. Please revise the entire manuscript in this regard, beginning with the Abstract. It is very clear that is “yours” paper.

Response to comment 1: Thank you for your comment, we revised our manuscript regarding this comment (Lines 76-84). Also, we performed English editing through professional editing service. 

Comment 2: No need to underline the first letter of the words in L74, L108, L115, etc. Please remove the underline in the entire manuscript.

Response to comment 2: As suggested, we removed underlines as you commented (Lines 77-78, 111-112, and 120). 

Comment 3: In 2. Materials and Methods section, I recommend a subsection to be added, related to the criteria of papers selection. In this regard, a PRISMA flow chart is recommended, according to Page, M.J.; et al. The PRISMA 2020 statement: An updated guideline for reporting systematic reviews.

Response to comment 3: As suggested, we found that description about functional classification via literature search was ambiguous. To clarify this section, we revised the description (Lines 114-119) and added information of the functional characterized genes as supplementary table S1.

Comment 4: L112. EXCEL which version? 

Response to comment 4: We added version information of EXCEL software in the manuscript (Lines 115-116).

Comment 5: Discussion part must be extended, it is much too short. As the title of the paper mentions “… to Facilitate Productivity” in the Discussion section it must be mentioned a paragraph related to the ways of increasing production taking into account the soil properties, crop rotation, climatic aspects and the use of fertilizers.

Response to comment 5: To address this comment, we revised the overall discussion section by adding descriptions for ways to improve productivity via adapting our results (Lines 300-340). 

Round 2

Reviewer 1 Report

The revised manuscript has much improved. I appreciate the authors' efforts to address the comments expressed during the previous round of reviews.

Reviewer 2 Report

All my requests have been addressed.